# Household perceptions, practices, and experiences with real-world alternating dual-pit latrines treated with storage and lime in rural Cambodia

James Harper[1]*, Rana Abdel Sattar[2], Tyler Kozole[2], Veasna Toeur[2], Jennifer Rogla[3], Marlaina Ross[4], Nate Ives[5], Hannah Pruitt[4], Payal Soneja[4], Drew Capone[6]

1 University of Colorado Boulder & Research Manager, Realize Research, LLC, Nederland, Colorado, United States of America, 2 iDE Cambodia, Boeung Tumpun, Cambodia, 3 iDE, Denver, Colorado, United States of America, 4 Causal Design, Phnom Penh, Cambodia, 5 USAID, Washington, DC, United States of America, 6 Department of Environmental and Occupational Health, School of Public Health-Bloomington, Indiana University, Bloomington, Indiana, United States of America

* james.harper@colorado.edu

**Editor:** D. Daniel, Gadjah Mada University Faculty of Medicine, Public Health, and Nursing: Universitas Gadjah Mada Fakultas Kedokteran Kesehatan Masyarakat dan Keperawatan, INDONESIA

## Abstract

Achieving universal safely managed sanitation (SMS) is an ambitious long-term goal in resource-limited rural areas. The non-governmental organization, iDE, introduced the alternating dual-pit latrine (ADP), which treats fecal sludge (FS) on-site using storage and lime to increase SMS in rural Cambodia. However, SMS via ADPs requires adherence to recommended practices (e.g., how and when to switch between pits). We surveyed 765 rural households with ADPs across five of 25 Cambodian provinces to understand how real-world household sanitation practices and knowledge affect and are related to adherence to recommended practices at scale. We calculated summary statistics of household survey responses and used regression models of composite indices to describe how households' practices and attitudes related to ADPs affect adherence to recommended ADP practices. By 24 months after training, three in five households did not recall how long treatment must proceed until emptying can be performed safely. No household waited the recommended two years to empty their pits. While households appreciated the advantages of owning an ADP (e.g., reduced costs and required land area compared to single-pit latrines over time), no household followed recommended treatment practices. This lack of adherence could have health and environmental implications for households using ADPs. Household practices also varied by province, flood proneness, and education level, adding complexity to how to improve adherence; for example, having at least one household member that completed formal education surprisingly reduced compliance with recommended ADP practices. Household behaviors impact the use and maintenance of on-site sanitation systems in rural areas, with proper adherence is necessary to achieve sustained SMS. Increased access to affordable and safe emptying

**Data availability statement:** The datasets generated and analyzed in this study, along with the code used for analysis, are available in the repository "Microbial Hazards and Household Perceptions, Practices, and Experiences with Real-world Alternating Dual-Pit Latrines Treated with Storage and Lime in Rural Cambodia" at https://osf.io/uwq82/.

**Funding:** Grant funding provided by Australian Aid, Water for Women, USAID, and The Stone Family Foundation. The funders had no role in study design, data collection and analysis, decision to publish, or preparation of the manuscript.

**Competing interests:** The authors have declared that no competing interests exist.

service providers could enable households to manage the pits of their ADPs and dispose of FS in-situ according to treatment duration, while also ensuring that household practices in operating on-site sanitation systems are integrated into the design, installation, and SMS monitoring of such systems.

## 1. Introduction

Safely managed sanitation (SMS) prevents exposure to enteric pathogens by promoting safe and effective fecal sludge management (FSM), which ensures that human feces and urine (fecal sludge, FS) are contained, transported, treated, and disposed of or reused in situ or offsite [1–4]. Safely managed on-site sanitation (SMOSS) systems treat FS where it is contained, which provides multiple benefits: 1) FS does not need to be transported for treatment or disposal; 2) there are fewer opportunities for people to be exposed to the pathogens in untreated FS; and 3) in situ FS treatment by household owners is typically cheaper and less complex compared to off-site treatment systems [5–7]. Thus, SMOSS systems provide an alternate path through the typical sanitation value chain and solve many common problems of off-site treatment [5,6]. Examples of SMOSS systems include alternating dual-/twin-pit latrines (ADPs), composting toilets, and septic systems but can also include single-pit latrines that are simply capped when full and a new pit constructed [8].

Due to their many benefits, tens of millions of SMOSS systems have been installed globally, particularly in rural areas. For example, millions of ADPs were recently constructed by India's Swachh Bharat Mission [9], and various programs have built millions of SMOSS systems in Bangladesh, Ecuador, Indonesia, Kenya, Serbia, Zambia, and Cambodia [6,10–12]. As highlighted by the World Health Organization (WHO) and others, ADPs are by far the most common rural SMOSS systems constructed globally – they are simple to operate, have purportedly high treatment effectiveness, and are low cost compared to other SMOSS system [8,13,14].

However, ADPs have not yet been studied at scale in a real-world setting to assess whether their use and treatment effectiveness are as expected for the millions of households that now own them. As the Joint Monitoring Programme (JMP) states, accurate data that describe whether ADPs *actually achieve* SMS are not consistently available [6]. ADPs' treatment effectiveness is dependent on individual households' operating practices; therefore, understanding how households operate ADPs in the real world can inform behavior-change campaigns that use techniques like community engagement, tailored trainings, peer education, and positive deviance to encourage households to operate their ADPs as recommended [8,15]. Lastly, it has been recently shown that climate events like flooding can reduce the safety of household sanitation practices [16]. In Cambodia, flooding is typically the most common and influential climate hazard [17–20]; thus, we focus on flood-proneness, which is described by the World Health Organization to refer to any land area that is susceptible to inundation by water from any source that poses a risk to human health and well-being [19]. Thus, improving public understanding of sanitation practices,

including ADP operation, can enable communities to select appropriate practices for their locations and mitigate health-related climate risks.

To address this gap in the literature, this study investigates how households operate ADPs in rural Cambodia and the effect of those practices on the treatment effectiveness of ADPs. Cambodia provides an excellent case study due to the pressing need for widespread sanitation solutions: basic sanitation coverage – "the percentage of improved sanitation systems in a region that are used by only one household and hygienically separate human excreta from human contact" [6] per the United Nations' Joint Monitoring Programme – in rural Cambodia has rapidly increased in recent years, and most rural Cambodian households now have a pour-flush single-pit latrine [12]. Although these systems effectively capture and contain FS, they likely will not result in sustained SMS because 1) little to no FSM infrastructure or enforced regulations that require FS to be treated, disposed of, and/or reused safely exist; and 2) few pit-emptying service providers are available and affordable [6,21]. Thus, rural Cambodian households typically empty their own untreated FS, posing a serious health risk [12]. In response, the Cambodian national government and sanitation sector recommend SMOSS systems in rural areas, treating FS on-site and using treated FS as fertilizer [22]. For example, non-profit organizations like International Development Enterprises (iDE), Stichting Nederlandse Vrijwilligers (SNV), and WaterSHED have designed and marketed SMOSS systems across rural Cambodia [23]. iDE has recently scaled up their sanitation-marketing program to also include upgrading existing single-pit latrines to ADPs with lime treatment, making households with their products ideal for study [24].

Inspired by the traditional alternating twin-pit design [25], the ADP provides rural Cambodian households with context-appropriate modifications, including lime treatment, a visible gauge that signals when a household's pit is nearly full, and the option to buy an ADP as an upgrade to an existing single pit latrine [22,24,26]. An ADP treats FS on-site while maintaining latrine functionality when a pit fills and effectively doubles FS storage capacity [24]. At a lower cost than many alternatives, an ADP also requires less land area than multiple single-pit latrines over time; this space-saving feature of ADPs is strongly preferred by rural Cambodian households [24,27]. Appropriate installation and operation of ADPs are critical to achieving sustained SMS using these SMOSS systems, and to support appropriate ADP installation and operation, iDE provides a list of recommended practices that households should follow to achieve sustained SMS with their ADPs.

As recommended by iDE, an ADP must be constructed by a trained installer that follows specific steps to ensure FS containment and treatment, including proper site selection, disconnecting the toilet from the existing pit, and mixing 6 kg of hydrated lime into the existing pit's FS (S1 Fig and S1 File). The installer must then construct a new pit and connect it to the toilet, which can be used immediately. After ADP installation, the household is instructed on how to safely operate the ADP and given a leaflet with clear operating instructions (S2 Fig). These recommended practices include 1) waiting for at least two years before emptying a pit undergoing treatment to allow pathogens to inactivate, reducing the risk of disease; 2) emptying the treated FS in the disconnected pit; and 3) having a trained service provider treat the full pit with lime and disconnect the full pit from the toilet and reconnect toilet to the empty pit. For its end use, emptied FS can be reused as fertilizer, as recommended by the World Health Organization and the National Government of Cambodia [14,22]. More details about recommended ADP installation and operational practices can be found in iDE's ADP Installation Manual and Technical Guidelines in Supporting Information S1 File.

With more than 23,000 rural Cambodian households using ADPs since 2017, the effectiveness of ADPs treating household FS and thereby achieving sustained SMS can be evaluated. In 2021, a large sample of these households were studied to understand both household behavior and microbial hazards in this real-world setting. Results describing the microbial hazards of emptying real-world ADPs are described in a companion article [28] and highlight that emptying an ADP whose FS was treated with lime and stored for the WHO-recommended two years poses a one-in-three risk of exposure to microbial hazard via *Escherichia coli* (*E. coli*) and fecal coliforms. That study also highlighted that many ADPs were not operated as recommended, and that additional research should investigate household ADP behaviors. Those results provide appropriate background and add to the comprehensive view of FS treatment with ADPs, to which this

study adds. In this study on household behavior, we investigate the question "How do household-level factors affect the ADP's potential to achieve SMS?". We hypothesize that, despite the potential for ADP technology to safely treat FS over time, ADPs are limited in their effectiveness at achieving SMS because households may not adhere to recommended practices.

## 2. Methods

### Study design, sampling frame, inclusion criteria, and sample size

To maximize generalizability, a random sample of rural ADP-owning households was drawn from iDE's historical sales data in five Cambodian provinces where ADPs were sold at scale. From the random sample, a phone survey then identified households with filled pits to ensure that households included in the study had at least an opportunity to treat FS and switch pits. Households were only eligible for inclusion in the study if they owned their ADP for more than two years and had switched their pit connections from one full pit to the other at least once since their ADP was installed (full inclusion criteria listed in the Supplementary Information S2 File). With 8,410 ADPs used for at least two years per iDE's sales database and an estimated 25% of ADPs filling after two years per the phone survey, 765 ADP-owning households were surveyed (S1 Table) [29,30]. Following standard practice described in the literature and internationally accepted protocols, the study design aimed to achieve a 5% precision when describing the target population [31]; however, practical restrictions due to accessing respondents that met all inclusion criteria, travel distances, and budget limited the survey's precision to 6%. A description of the study population, including household demographics and details about household latrines is available in Supporting Information S3 File. This study design was approved in the Solutions IRB protocol 2021/10/2.

### Data collection

A 40-minute in-person survey (see Supporting Information A) was developed and administered to 765 rural households in five of 25 Cambodian provinces where ADP installations were sold at scale. Based on a review of existing literature, including reports from WASH-sector organizations in Cambodia to ensure contextual relevance, the topics included in the survey were evidence-backed factors that affect sanitation practices and included:

- *Demographics*: household size, highest education level of any household member, primary occupation or livelihood activities of household members, poverty status (IDPoor), gender of respondent, household members with disabilities, and ethnicity. Household poverty levels were self-reported using the Cambodian National Government's Identification of Poor Households (IDPoor) Programme metrics [32];

- *Sanitation history and toilet use information*: number of latrine pits, frequency of use, number of people using the latrine, sharing status, improvements, system modification (e.g., a pit is "pierced"; i.e., an overflow hole was installed in a pit) and maintenance issues (e.g., repairs);

- *Knowledge of ADPs and SMS*: understanding of FS dangers; pit-emptying and treatment processes that can improve safety; equipment and clothing used to empty pits safely; and methods of emptying pits;

- *ADP experience*: how, when, and why latrine was switched to a different pit; perceptions of expense, disgust, difficulty, and safety; disposal location; method of disposal; and maintenance issues (e.g., repairs); and

- *Attitudes about ADP and SMS*: perceptions of ADPs, including safety, ease of use, maintenance, disgust, and affordability.

All interviews were conducted at the site of the household-owned ADP using secure smartphones, and data was managed and stored on secure servers. The study design was approved in the Solutions IRB protocol 2021/10/2, and respondents provided oral informed consent to participate prior to beginning each interview. Written consent could not be obtained because

most study participants were illiterate; Solutions IRB approved the use of oral consent; and oral consent was documented digitally, along with all other responses, for each household by recording the response of each study participant to the question "Do you consent to being a part of this study?". Data collection occurred between Mar 31, 2022 and May 4, 2022.

## Data analysis

Survey data were analyzed to describe 1) households' knowledge and perceptions of recommended ADP practices and sanitation in general; 2) households' practices and experiences with ADPs; and 3) common characteristics of households that use ADPs as recommended. Response frequencies were used to analyze these aspects of households, and descriptive statistics were disaggregated by province and geographic area (flood-prone vs. non-flood-prone), where relevant. To identify common characteristics of households that use ADPs as recommended (S2 Fig), five multi-factor indices were developed to describe households' practices and perceptions related to recommended practices for ADP operation. These five ADP Indices are:

1. The **Emptying Practices Index** describes whether households emptied their ADP's pits as recommended, including whether the household ever emptied a pit; whether the pit was emptied by a service provider; whether the pit was emptied using recommended practices; if the pit was pierced; whether the pit was left disconnected for a sufficient period of time before being emptied; and whether FS was disposed of in a recommended location and using recommended methods.

2. The **Treatment Practices Index** describes whether households treated FS stored in their ADP as recommended, including whether the household treated the new pit with lime before switching pits; whether lime treatment was performed by a service provider; and whether any product was used to treat waste in the pit before or after emptying.

3. The **Switching Practices Index** describes whether households switched their toilet from one pit to another as recommended, including whether the old pit was emptied after the household changed their connection to the new pit; whether the old pit was reconnected only after being disconnected for at least two years; whether the connection was changed by a service provider; and whether the old pit was disconnected from the toilet after switching pits.

4. The **Sanitation Knowledge Index** describes households' understanding of recommended ADP practices, including whether the household understands that sanitation affects public health; and that disposing of FS into a body of water or onto a field is unsafe.

5. The **Sanitation Attitudes Index** describes households' perceptions of the importance of following recommended ADP practices, including whether the household feels that treating FS is important; that safety is a priority when disposing of FS; and that it is appropriate for a household member to empty a pit with untreated FS.

The survey questions used to create each index are shown in S2 Table, and additional details are shown in S3 Table. Each survey question that was used to calculate each index was selected for that index based on decades of existing literature and iDE's knowledge of the local context in rural Cambodia [21,27,33,34]. Considering the Emptying Index as an example, all survey questions used to calculate this index specifically addressed the emptying-related topics listed above, which had been reported in the literature to affect or be related to pit emptying. For example, the question "Did the household pierce their pit?" was included in the Emptying Index because many past studies have shown that pit piercing is a method commonly used by rural Cambodian households to keep their pits from filling and thereby delay emptying. Other indices were constructed similarly, and additional rationale for including each survey question in their respective index is included in S3 Table. Survey questions that had not been specifically related to an index in the literature was not included in the calculation of an index. Also, survey responses that showed correlation above 0.2 (e.g., poverty and education) were not included in any index to avoid collinearity.

To calculate each index, the responses to each question were scored on a three-point numeric scale that categorized the related practice, knowledge, or attitude as "contrary to recommended ADP practices" (0), "neither contrary to nor following recommended ADP practices" (1), and "following recommended ADP practices" (2). For questions with binary response options, responses were scored as "contrary to recommended ADP practices" (0) or "following recommended ADP practices" (2). The final score for each index was calculated as the average score for each response, which also standardized the indices. Possible scores on each index ranged from 0 to 2.

With no strong rationale or data-driven evidence to suggest that certain survey questions were inherently more important to their respective indices than others, the scores for each survey question were not weighted when creating each index to maintain transparency and objectivity of the resulting composite indices, as described in the literature [35–37].

Each index was used as a dependent variable in generalized linear regression models coded in R (version 4.2.3) to describe the relationship between households' characteristics and these aspects of ADP operation. Household characteristics included in these models were geographic location, flood-proneness, education, poverty status, education, and pit overflow frequency (the number of times that FS exits a pit untreated) [32,38].

## 3. Results and discussion

### Study population

Approximately two thirds of households (540) were located in an area identified as flood-prone. The majority of study communities were located in close proximity to water bodies (e.g., lakes and rivers), which is common in rural Cambodia and highlights the ubiquity of flood-proneness for the majority of ADP customers [12]. Nearly two thirds of respondents (63%) were women, which likely occurred because men were more likely to be working outside of the household when surveying occurred [27,33]. Only 6% of households in the study population were identified as poor (either IDPoor 1 or IDPoor 2) [32], and two thirds of households reported that at least one household member had completed secondary school. More households were located in Svay Rieng and Prey Veng provinces, and the fewest were located in Kandal province.

The latrines of households in the study population were used by an average of 8.2 people daily. A majority of households did not share their latrine with anyone outside of their household (63%), and the average number of latrine users within these households was 6.2. The remaining 37% of households shared their latrine with, on average, 5.3 individuals from outside of the household. Prior sanitation ownership was high, with nearly all households having at least one pit prior to ADP installation (99%).

More details about the study population are available in Supplementary Information S3 File.

### Household perceptions and knowledge of ADPs

While nearly four-in-five households valued the advantages provided by ADPs compared to a single-pit latrine (78%), which included their reduced cost and required land area over time (23%); their doubled FS storage capacity (16%); and continuous latrine use (10%), only 13% specifically valued the FS treatment provided by ADPs. This finding is supported by low overall levels of household knowledge about ADPs: only one in seven households understood that they had to treat each pit of FS with lime once it fills, and only one in eleven households understood that treated FS could be emptied safely after two years of storage treatment. Nevertheless, most households perceived emptying untreated FS to be unsafe (82%) and considered FS treatment and safe disposal locations to be important and to foster good relationships with neighbors (97%). Thus, there is a disconnect between households' valuation of ADPs when it comes to their attitudes around SMS and how they actually operate their ADPs to achieve sustained SMS.

### Household ADP practices compared to recommended ADP practices

No households followed all recommended ADP practices. With 64% emptying their pits themselves, no household waited the recommended two years to empty their pit undergoing treatment, risking direct exposure to untreated FS; only one in

 

eleven waited longer than a year and a half; and six in ten waited less than one year. Regarding lime treatment, only 8% of pits were treated with lime by a trained service provider after switching pits. Although another 6% of pits were treated with lime by a household member, households were advised to contract a trained service provider to do this and therefore not trained to treat FS with lime and likely did not perform this process correctly. The most common reasons for not treating FS with lime were a lack of required materials, primarily lime (33%), and a lack of available service providers (23%).

Pit-switching practices overall did not follow recommended practices and varied widely: only 14% of households switched pits after emptying their disconnected pit; other households performed a variety of other practices that were either unsafe or did not take advantage of the ADP's design, including emptying their connected pit *immediately* (not after storage treatment) themselves (31%) or hiring a service provider to empty it (24%); piercing their pit to maintain latrine functionality (15%); or switching pits *without emptying* the disconnected pit (14%) as the disconnected pit typically had additional capacity available given normal dewatering that occurred after it was disconnected. Switching pits without emptying the disconnected pit is unsafe because it exposes the FS undergoing treatment to untreated FS, effectively negating all treatment in the ADP [28].

## Factors affecting households' ADP practices and sanitation knowledge and attitudes

Poor sanitation knowledge often precludes performing safe sanitation practices [2,27], providing one potential explanation for the unsafe household behaviors observed in the sample. Households' performance of recommended ADP practices and their sanitation knowledge and attitudes also vary due to many other factors. For example, the literature highlights that convenience and cost avoidance are strong motivators for households when managing FS [27,39,40]. For example, most households choosing to either 1) empty their connected pit immediately upon filling without any FS treatment, or 2) piercing their pit to prevent it from filling, exemplify practices of convenience and cost avoidance because they can be performed by households without a service provider. Lastly, households may not follow recommended practices because these practices are simply difficult to understand or implement.

Disaggregating household characteristics by ADP index allows us to identify what factors affect each step in managing FS with an ADP (Table 1 and S4–S8 Tables).

Completing formal education was associated with *less frequent* performance of recommended ADP emptying and pit switching as well as worse sanitation knowledge. Households with a member that had completed formal primary and secondary education surprisingly performed recommended emptying practices less frequently; this result disagrees with the literature, which reports that more education, particularly attaining at least a primary education, is associated with performing safer sanitation practices [34]. This discontinuity can likely be explained by the different methods used in this study

Table 1. Effects of household characteristics on ADP indices describing ADP practices and sanitation knowledge and attitudes[a].

| Household Characteristic ADP Index | Completing formal education | Poverty | Frequency of pit-overflow events | Flood proneness | Province |
|---|---|---|---|---|---|
| Emptying | – | | – | | • |
| Treatment | | + | + | – | |
| Switching | – | – | | | |
| Sanitation Knowledge | – | + | – | | • |
| Sanitation Attitudes | | | | – | • |

[a]: "-"indicates a lower ADP index value (i.e., less frequent performance of recommended ADP practices, less knowledge of SMS, worse attitudes towards SMS) as the household characteristic's value increases (e.g., the "-"under Education and across from Emptying indicates that more education decreases the performance of recommended ADP emptying practices.).

Conversely, "+" indicates a higher ADP index value as the household characteristic's value increases. "•" indicates the ADP index varies with location. A blank cell indicates no association between recommended ADP practices and the household characteristic.

(an index that considers multiple facets of SMS and removes confounders) compared to that in the literature (correlations between individual metrics). Also, households with no formal education may only learn about sanitation and associated topics (e.g., germs, public health) from the recommended ADP practices they are taught when their ADP is installed, reducing any possible confusion or conflict with other previous knowledge sources. The completion of formal education in rural Cambodia is not universal; those who do complete schooling may be more confident in thinking for themselves and thus inclined to perform sanitation process, like pit switching, themselves. This behavioral construct is described in the literature by the concept of perceived control [41] and may help explain why formal education and sanitation knowledge correlate with lower levels of recommended ADP emptying and pit-switching practices. Similarly, having vocational education was associated with a slight decrease in recommended switching practices.

Poverty was associated with more frequent performance of recommended treatment practices and better sanitation knowledge, but less frequent performance of recommended switching practices. Additional research is needed to explain these associations.

A higher frequency of pit-overflow events was associated with more frequent performance of recommended treatment practices but less frequent performance of recommended emptying practices and worse sanitation knowledge. First, the connection between more frequent pit-overflow events and worse sanitation knowledge is self-evident: households with poor sanitation knowledge are more likely to not empty their pit before it overflows because they likely do not understand the effects of such inaction [5,21]. Considering emptying practices, pits commonly overflow in rural Cambodia due to rainfall and flooding [20,42]; as a result, households frequently empty their pits to maintain latrine functionality by piercing them [43], which is not a recommended ADP emptying practice. Also, pit-overflow events require households to manage FS, regardless of the practice used; thus, as the literature states, households under these climatic conditions are likely to become more comfortable and familiar with managing FS, which can encourage them to perform whatever practices are most convenient or cost-avoidant, which are frequently unsafe [44]. Lastly, the increase in performance of recommended treatment practices with increasing pit-overflow events may occur due to households' well-documented desire to "kill *meruk*" (meaning desire to reduce pathogen concentrations in FS) and prevent it from entering the environment, which likely leads them to treat FS following recommended practices given the number of times they must manage these events [27].

Flood proneness was associated with less frequent performance of recommended treatment practices and worse sanitation attitudes. These results support the literature, which states that flood proneness increases challenges related to latrine functionality and achieving SMS, and is typically associated with unsafe FSM practices [16].

Households in different provinces tended to operate their ADPs in different ways. For example, households in Prey Veng and Svay Rieng followed recommended emptying practices less frequently and had worse sanitation attitudes compared to the reference province Kampong Thom, while households in Siem Reap had more sanitation knowledge than the reference province Kampong Thom. Supporting the literature [21,45], these results highlight that regional variations in ADP practices and sanitation knowledge and attitudes are large and must be considered when developing ADP educational materials and targeting related behavior change campaigns.

## Limitations

The results of this study are subject to some important limitations. Social desirability bias likely occurred during surveying. Particularly given questions about sensitive or taboo topics like sanitation, respondents tend to provide answers that they view as more socially acceptable but perhaps less accurate. This bias was mitigated by a brief discussion between enumerators and participants about why talking about such topics is important. All participants were also reminded that they did not have to give a response to any question.

Similarly, recall bias, which describes how respondents may remember past events inaccurately, likely affected responses in this study, particularly regarding whether and how lime was applied to a pit nearly two years prior. Recall

bias was minimized by shortening recall periods, using cued recalls, and using landmark events (e.g., the rainy season) instead of calendar dates when possible throughout the survey.

The well-documented lack of service providers in rural Cambodia likely also affected responses to questions that were used to calculate the Emptying Practices index [21,33,46]. For example, households likely emptied their pits themselves (i.e., did not hire a trained service provider) with buckets or pumps (i.e., not with a safer vacuum truck) and pierced their pits more frequently due to a lack of ability to hire a trained service provider to empty their pit using safer methods. Thus, the availability of service providers, which varies across different districts and provinces, likely affected the results of this index and the results reported with this index [33].

Lastly, a systematic behavior bias was introduced into the study design by the sampling frame. The selected sample specifically included only ADP customers who had owned their ADPs between two and four years, and who had managed FS in some way (e.g., emptied their pit, switched their pit). Therefore, this study may describe a group of people that are different from the typical Cambodian rural population. Household demographics and an appropriate sample size and study design were used to ensure that the sample population matched the target population of all rural households in Cambodia; however, unmeasured or unknown data may have introduced a systematic bias that cannot be detected.

## 4. Conclusions

Although another study has shown that the ADP technology can effectively treat FS [28], this study showed that household adherence to recommended ADP practices was low, reducing the prospect of achieving sustained SMS. Emptying was performed before the recommended two years of storage treatment [14]; lime treatment processes were not performed as recommended and varied with flood proneness and poverty; and pit switching was performed in the wrong sequence or contaminated FS undergoing treatment. Although households expressed that they value ADPs' advantages, their operation of ADPs decreased ADPs' potential of achieving sustained SMS.

Results from this study highlight that household knowledge, education, and experience *do not necessarily* lead to safer sanitation practices. Education did not positively correlate with performing recommended practices; other motivations like convenience, cost avoidance, and comfort likely play stronger roles and must also be addressed to attempt to achieve SMS with ADPs. Also, the effects of poverty, experience managing FS in the past, and the social norms in different geographies play key roles in household ADP practices and must be addressed systemically to allow for the potential for ADPs to achieve SMS.

The results of this study allow us to propose pathways forward to improve sanitation processes to help achieve sustained SMS in rural Cambodia. First and foremost, there is a critical need for inclusion of questions about user maintenance within monitoring frameworks when assessing SMS. While data is often available for urban centralized wastewater treatment of sewage, marked data gaps remain to accurately monitor SMOSS systems [6,47]. This study showed that households rarely follow recommended ADP practices, which can prevent achieving sustained SMS. Monitoring of key performance indicators related to households operating ADPs as recommended must be used to evaluate the holistic effectiveness of SMOSS systems like ADPs to ensure that FS is *actually* being safely managed. Another need identified by the JMP is the combination of data from both service providers and households, which is particularly relevant to service-dependent SMOSS systems, such as ADPs. Indicators monitoring SMOSS systems must therefore go beyond the on-site storage and treatment capacity of the system to include household- and service-provider-level data that describe the operation of those systems to achieve sustained SMS.

It is becoming increasingly possible to monitor containment, treatment, and disposal in-situ by adding core questions to routine household surveys and conducting household sanitary inspections [47]. These follow-up visits with ADP-owning households, if made at regular intervals, can provide timely monitoring data and will also serve to remind households of how to operate their ADPs to achieve sustained SMS. These additional touch points would likely also improve households' ADP-related knowledge retention over the long term and may improve households' adherence to recommended ADP practices. However, there is still a need for more affordable, practical SMS monitoring tools that evaluate household

sanitation knowledge, preferences, and practices, in addition to sanitation system functionality and indicators of unsafely managed sanitation (e.g., open discharging pits) to appropriately gauge progress towards achieving sustained SMS.

Relatedly, SMOSS systems' reliance on households operating them in specific ways requires that they be designed with households' needs and preferences as critical factors. If a SMOSS system is not perceived as valuable, affordable, or feasible to maintain appropriately by households, then it will not be. This study highlights that convenience and value are key design features; thus, considerable investment must be made in identifying key household needs using user-centric design processes, including human centered design and focused market research. Then, SMOSS systems, including their technology and supporting service delivery, must be designed around these needs. With these considerations in mind and based on the results of this study, future ADP delivery could be improved by 1) systematic latrine-supplier follow-up visits that offer households a pit-switching-and-lime-treatment service; 2) community-wide local authority engagement and behavior change campaigns that directly target achieving sustained SMS; 3) offering various on-site SMS products that meet various community needs (e.g., on-site emptying services); and 4) adopting an evidence-based intervention model that allows for design iterations, research, and testing to monitor, evaluate, and improve the impact of SMOSS systems.

The need for improved access to climate-resilient SMS, particularly in flood-prone areas, is highlighted in this study. While ADPs have the potential to achieve sustained SMS if operated as recommended, they are not appropriate in all settings. The effects of climate hazards, most notably flooding, on sanitation systems must be considered during the design and construction of any sanitation system [16]. This study shows an association between flooding and households less frequently using recommended treatment practices and having worse sanitation attitudes. Therefore, SMOSS systems that are not designed to withstand flooding should not be installed in flood-prone regions. Considering the widespread likelihood of households emptying their full pits before the recommended two years, alternative on-site treatment processes that provide quicker, safe FS treatment should be made available to rural communities, especially in high groundwater and flood-prone areas where pits fill up more quickly; many such practices are described in the literature [8,28,48]. Also, determining the appropriate type of sanitation infrastructure for a given situation and setting should be improved by developing more practical contextual tools, including those that detect and consider flood proneness and perhaps broader climate vulnerability based on the local context [16].

In challenging environments, climate-resilient sanitation systems that provide preliminary treatment should be considered. These systems may include, for example, the Wetlands Work! HandyPod, which uses a microbial biofilm process to treat black-water effluent [49], and iDE and EWB Australia's All-Season Upgrade, which provides gravel-media filtration and a leach field to increase infiltration in high groundwater and dense soils [50]. However, even with more of these climate-resilient sanitation systems installed, FS emptying, treatment, and disposal solutions are needed to achieve sustained SMS. These solutions may include on-site trenching and burial of FS during the dry season and off-site conveyance to decentralized FS treatment plants.

As evidenced by the marked improvement of latrine coverage in rural Cambodia in recent decades [12], rural Cambodians value containing FS. However, the results of this study indicate that SMS has not yet been widely prioritized. It is unlikely that all households will start operating ADPs as recommended, even with widespread education and behavior change campaigns. Supported by the result that over 1 in 5 respondents reported not treating their pit with lime due to lack of access to service providers, it can be inferred that increased access to SMS services could lead to more widespread adoption of SMS practices. Improved access to affordable, quality SMS service provision would alleviate household responsibility for FSM and mitigate the common practice of self-emptying, where households typically expose themselves or their communities to untreated FS [21,27,33,46].

While formal education was not positively correlated with performing recommended ADP practices likely due to other motivations like convenience, cost avoidance, and comfort playing stronger roles, formal education may still play a critical role in achieving sustained SMS in rural Cambodia. While information specific to sanitation technologies like ADPs may not be practical to teach in formal local youth education settings, topics directly related to SMS including germ theory, pathogen inactivation via different treatment processes, and the safe reuse of household FS after effective treatment should be integrated into formal primary and secondary education curricula in rural Cambodia. Greater understanding in

rural Cambodia of how FS affects public health may provide a foundational underpinning for households to operate ADPs as recommended to achieve sustained SMS. Such education should take advantage of rural Cambodians' desire to "kill *meruk*" and be adjusted to consider observed regional variations in household sanitation knowledge, common deviations from recommended practices, and poor sanitation attitudes, as reported in this article.

To achieve sustained SMS in rural Cambodia and thereby improve public health, household perceptions, practices, and experiences must be understood and considered when designing on-site sanitation systems. Current SMS interventions and monitoring methods should continue being evaluated and improved to ensure that Sustainable Development Goal 6 is accomplished.

## Supporting information

**S1 File. ADP installation manual and technical guidelines, behavior survey, and collected data.**
(DOCX)

**S2 File. Inclusion criteria.**
(DOCX)

**S3 File. Description of the study population.**
(DOCX)

**S1 Fig. Product details of alternating dual-pit upgrade provided by iDE to rural latrine installers in Cambodia.**
(TIF)

**S2 Fig. iDE-recommended household SMS practices after the purchase and installation of an alternating dual-pit latrine.**
(TIF)

**S1 Table. Sample size calculation.**
(DOCX)

**S2 Table. Five ADP Indices with the survey questions used to construct them.**
(DOCX)

**S3 Table. Five ADP Indices with associated questions, possible responses, scores, and score rationales.**
(DOCX)

**S4 Table. Linear regression results of the Emptying Practices Index.**
(DOCX)

**S5 Table. Linear regression results of the Treatment Practices Index.**
(DOCX)

**S6 Table. Linear regression results of the Switching Practices Index.**
(DOCX)

**S7 Table. Linear regression results of the Sanitation Knowledge Index.**
(DOCX)

**S8 Table. Linear regression results of the Sanitation Attitudes Index.**
(DOCX)

## Acknowledgments

The authors would like to thank iDE Cambodia for leading the design and implementation of this study; iDE Global staff for providing expert feedback in WASH and evidence and analytics; Causal Design Inc. for managing all aspects of the study; Andy Robinson for his continuous feedback throughout the study and reviewing this article; Water WISER and Cranfield University, specifically Alison Parker, Francis Hassard and Barbara Evans, for their feedback on the preliminary study design and research questions; Joe Brown for guidance on the study design, specifically regarding the biology of ADPs; Ronald Beckett and Katherine Harper-Beckett for childcare that enabled the focused time and energy needed to complete this manuscript; and Mimi Jenkins for support and encouragement with the genesis of this research initiative.

## Author contributions

**Conceptualization:** James Harper, Rana Abdel Sattar, Tyler Kozole, Veasna Toeur, Jennifer Rogla, Drew Capone.

**Data curation:** James Harper, Rana Abdel Sattar, Tyler Kozole, Veasna Toeur, Marlaina Ross, Nate Ives, Hannah Pruitt, Payal Soneja.

**Formal analysis:** James Harper, Hannah Pruitt, Payal Soneja.

**Funding acquisition:** Rana Abdel Sattar, Tyler Kozole, Veasna Toeur.

**Investigation:** James Harper, Nate Ives, Drew Capone.

**Methodology:** James Harper, Marlaina Ross.

**Project administration:** James Harper, Rana Abdel Sattar, Veasna Toeur, Jennifer Rogla, Marlaina Ross.

**Resources:** James Harper, Marlaina Ross.

**Software:** James Harper.

**Supervision:** James Harper, Rana Abdel Sattar, Tyler Kozole, Veasna Toeur, Jennifer Rogla, Marlaina Ross, Nate Ives.

**Validation:** James Harper, Rana Abdel Sattar, Tyler Kozole, Marlaina Ross.

**Visualization:** James Harper.

**Writing – original draft:** James Harper, Payal Soneja.

**Writing – review & editing:** James Harper, Rana Abdel Sattar, Tyler Kozole, Veasna Toeur, Jennifer Rogla, Marlaina Ross, Nate Ives, Hannah Pruitt, Payal Soneja, Drew Capone.

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
