## [Decision Letter · Decision Letter 0]

26 Nov 2024

PONE-D-24-32665Household Perceptions, Practices, and Experiences with Real-world Alternating Dual-Pit Latrines Treated with Storage and Lime in Rural CambodiaPLOS ONE

Dear Dr. Harper, PE, PhD,

Thank you for submitting your manuscript to PLOS ONE. After careful consideration, we feel that it has merit but does not fully meet PLOS ONE’s publication criteria as it currently stands. Therefore, we invite you to submit a revised version of the manuscript that addresses the points raised during the review process.

**ACADEMIC EDITOR: ** Even though one reviewer suggests to reject your manuscript, I think authors can still improve the draft. Please respond carefully to the reviewers' comments.

We look forward to receiving your revised manuscript.

Kind regards,

D. Daniel, Ph.D.

Academic Editor

PLOS ONE

Journal Requirements:

2. Thank you for stating the following financial disclosure: Grant funding provided by Australian Aid, Water for Women, USAID, and The Stone Family Foundation. 

4. We notice that your supplementary figures and tables are included in the manuscript file. Please remove them and upload them with the file type 'Supporting Information'. Please ensure that each Supporting Information file has a legend listed in the manuscript after the references list.

Reviewers' comments:

Reviewer's Responses to Questions

**Comments to the Author**

1. Is the manuscript technically sound, and do the data support the conclusions?

Reviewer #1: Yes

Reviewer #2: No

2. Has the statistical analysis been performed appropriately and rigorously? 

Reviewer #1: Yes

Reviewer #2: No

3. Have the authors made all data underlying the findings in their manuscript fully available?

Reviewer #1: Yes

Reviewer #2: Yes

4. Is the manuscript presented in an intelligible fashion and written in standard English?

Reviewer #1: Yes

Reviewer #2: Yes

5. Review Comments to the Author

Reviewer #1: The manuscript has been scientifically written evident with the data provided and the conclusion has been drawn after relevant statistical inference has been employed and justified. The statistical inference employed in this study answers the objective of the study assisted by the rigorous data collection techniques used in this study which was attached too. The layout and presentation of the study show a thorough cohesion throughout and is written in a manner that the reader can comprehend. With that said I recommend this submission for publication

Reviewer #2: This paper summarizes data collected during surveys of people using alternating dual-pit latrines (ADPs) across 5 rural provinces in Cambodia, a previously understudied safely managed sanitation system. Surveys collected information on sanitation management practices as well as knowledge and attitudes about ADP use. The authors used descriptive summaries of the data to assess the population’s sanitation practices and knowledge and developed indices summarizing various aspects of sanitation use to assess factors influencing sanitation practices, knowledge, and values. The authors conclude that the study population generally has poor sanitation knowledge and that reported ADP management consistently did not follow recommended practices. With results from the regression analyses, the authors further conclude that province, flood proneness, education, and other household factors impact people’s sanitation practices. My key concerns with this paper are that 1) there are issues with the quality and presentation of the methods and results sections, and 2) the conclusions do not align with the results presented in the paper. Specific concerns are outlined below:

Methods:

-There was no description of the study population or survey participants, including the age, gender, etc. A summary table outlining the demographics of the population that responded to the survey would have been helpful. In particular, it would be helpful to describe any key differences between provinces.

-It is unclear why a “6% precision” rate was chosen to determine the sample size. Please provide a rationale, and ideally a citation to back up this choice.

-In the way the authors developed the indices, all variables within each index are given the same weight, which may not be appropriate. Please provide a rationale for this decision and why a PCA or other type of factor analysis was not used to develop the indices.

-The definition of “flood-prone” was not provided, and conclusions from results using that variable were often conflated to discussing climate change as a whole. In the methods section, the author notes including “climate vulnerability” in the model, but it is unclear if that is referring to flooding risk or if other climate considerations were assessed. The author should clearly define what makes a household flood-prone vs. not flood-prone and describe why flood-risk was used as an indicator of climate vulnerability beyond other measures (i.e. risk of drought, water availability, etc.).

-It was unclear why the independent variables used in the regression model were chosen over others that were included in the survey. A theory of change diagram or other description would be helpful to understand why certain variables were included and not some of the other demographic factors listed in the survey. The independent variables seem at high risk of issues with multicollinearity (i.e. poverty and education) and overadjustment. Please provide a clear rationale for model development and variable selection. This could include correlation matrices to illustrate how the variables relate to one another in this specific sample.

Conclusions

-All conclusions about the regression model were drawn only based on p-values and no confidence intervals or estimates of variability were provided. This is not conventional and makes interpretation difficult. See, for example https://academic.oup.com/aje/article/186/6/627/4091005 and https://jamanetwork.com/journals/jama/fullarticle/2503156.

-Conclusions about the regression models do not accurately describe model estimates in terms of their reference category. For example, the author notes that "households in Prey Veng and Svay Rieng followed recommended emptying practices less frequently and had worse sanitation attitudes compared to other provinces". This statement suggests that models are comparing individual provinces to all other provinces, but the models include province on a categorical scale with Kampong Thom as the reference category. With this approach, it is not appropriate to draw conclusions about one province compared to all others, where regression coefficients are only comparing each province to the reference.

-Some conclusions made by the authors don’t seem to align with the presented data. For example, the author concludes that “households valued the advantages of ADPs”, when only a small percentage (at most 23%) reported valuing specific aspects of ADPs.

-The author should be careful not to conflate the issues of knowledge, attitudes, and practices in conclusions. For example, the author states that “households with poor sanitation knowledge did not highly value SMS and are therefore more likely to not empty their pit before it overflows”. Sanitation knowledge is not necessarily correlated with people’s values. It's very possible for people to not know a lot about sanitation and still value it.

-The conclusion section makes strong recommendations about the need for improved monitoring, but there does not seem to be clear evidence from the results of this paper pointing to that conclusion. Similarly, conclusions presented about the issue of service providers in the area are conflicting. In the results, only 23% of participants reported a lack of service providers as a key limitation to treating FS, but increasing the amount of service providers in the area is presented as a key recommendation, which does not seem to be supported by the results. Conclusions and recommendations should only be based on the data and scope presented in this paper.

Other general concerns:

-Results and discussion should be separated into two distinct sections, with the results section first presenting the objective results of the analyses (preferably using tables and figures to succinctly summarize findings), and the discussion section interpreting the results.

-It would be helpful to summarize findings from the companion article mentioned looking at microbial risks associated with ADPs. Additionally, details on the ADP management process were unclear and often confusing. The specific recommendations from iDE should be explained in detail in the introduction (not in a footnote), and it should be made more clear what procedures should be carried out by service providers and what should be done by households.

-It would be helpful to align the section headers in the methods and results sections to make it easier to follow the distinct research questions and analyses conducted.

6. PLOS authors have the option to publish the peer review history of their article (what does this mean? ). If published, this will include your full peer review and any attached files.

**Do you want your identity to be public for this peer review?** For information about this choice, including consent withdrawal, please see our Privacy Policy .

Reviewer #1: No

Reviewer #2: No

---

## [Author Response · Author response to Decision Letter 1]

9 May 2025

Please see attached file titled "Response to Reviewers". Thanks!

---

## [Decision Letter · Decision Letter 1]

21 Jun 2025

PONE-D-24-32665R1Household Perceptions, Practices, and Experiences with Real-world Alternating Dual-Pit Latrines Treated with Storage and Lime in Rural CambodiaPLOS ONE

Dear Dr. Harper, PE, PhD,

Thank you for submitting your manuscript to PLOS ONE. After careful consideration, we feel that it has merit but does not fully meet PLOS ONE’s publication criteria as it currently stands. Therefore, we invite you to submit a revised version of the manuscript that addresses the points raised during the review process.

**One reviewer is still requesting a minor revision. Please revise carefully.**

We look forward to receiving your revised manuscript.

Kind regards,

D. Daniel, Ph.D.

Academic Editor

PLOS ONE

**Journal Requirements:**

Reviewers' comments:

Reviewer's Responses to Questions

**Comments to the Author**

1. If the authors have adequately addressed your comments raised in a previous round of review and you feel that this manuscript is now acceptable for publication, you may indicate that here to bypass the “Comments to the Author” section, enter your conflict of interest statement in the “Confidential to Editor” section, and submit your "Accept" recommendation.

Reviewer #2: All comments have been addressed

2. Is the manuscript technically sound, and do the data support the conclusions?

Reviewer #2: Yes

3. Has the statistical analysis been performed appropriately and rigorously? 

Reviewer #2: Yes

4. Have the authors made all data underlying the findings in their manuscript fully available?

Reviewer #2: Yes

5. Is the manuscript presented in an intelligible fashion and written in standard English?

Reviewer #2: Yes

6. Review Comments to the Author

**Reviewer #2: ** This resubmission summarizes data collected during surveys of people using alternating dual-pit latrines (ADPs) across 5 rural provinces in Cambodia, a previously understudied safely managed sanitation system. The authors used descriptive summaries of the data to assess the population’s sanitation practices and knowledge and developed indices summarizing various aspects of sanitation use to assess factors influencing sanitation practices, knowledge, and values. The authors conclude that while ADPs have the potential to safely treat FS, low adherence to safe management practices by users may limit ADP’s ability be an effective technology for achieving safely managed sanitation in Cambodia.

With revisions to the manuscript, the authors appropriately addressed comments and improved the quality of the manuscript, particularly the description of the methods and interpretation of results. The authors provided important justification for model selection, including improved descriptions of variables and clarification about index weighting, that helps to validate their analysis approach. Additionally, the improved language in the discussion and results align more appropriately with the findings from the analyses.

There is one minor revision that would help provide important context for the reader - The information provided in supplemental file 3 is a valuable addition to this paper and would be helpful to include in the main body of the paper to inform the reader about important characteristics of the population. In particular, the summary on sanitation sharing and previous sanitation ownership included in S3 would be helpful to include in the main text to give readers insight into sanitation experiences in the study population. One key concern that should be addressed is that Table S3.6 is referenced in the supplemental file but does not appear to be included.

7. PLOS authors have the option to publish the peer review history of their article (what does this mean? ). If published, this will include your full peer review and any attached files.

**Do you want your identity to be public for this peer review?** For information about this choice, including consent withdrawal, please see our Privacy Policy .

Reviewer #2: No

---

## [Decision Letter · Decision Letter 2]

27 Aug 2025

Household Perceptions, Practices, and Experiences with Real-world Alternating Dual-Pit Latrines Treated with Storage and Lime in Rural Cambodia

PONE-D-24-32665R2

Dear Dr. Harper, PE, PhD,

We’re pleased to inform you that your manuscript has been judged scientifically suitable for publication and will be formally accepted for publication once it meets all outstanding technical requirements.

Kind regards,

D. Daniel, Ph.D.

Academic Editor

PLOS ONE

Additional Editor Comments (optional):

Reviewers' comments:

Reviewer's Responses to Questions

**Comments to the Author**

1. If the authors have adequately addressed your comments raised in a previous round of review and you feel that this manuscript is now acceptable for publication, you may indicate that here to bypass the “Comments to the Author” section, enter your conflict of interest statement in the “Confidential to Editor” section, and submit your "Accept" recommendation.

Reviewer #2: All comments have been addressed

2. Is the manuscript technically sound, and do the data support the conclusions?

Reviewer #2: Yes

3. Has the statistical analysis been performed appropriately and rigorously? 

Reviewer #2: Yes

4. Have the authors made all data underlying the findings in their manuscript fully available?

Reviewer #2: Yes

5. Is the manuscript presented in an intelligible fashion and written in standard English?

Reviewer #2: Yes

6. Review Comments to the Author

Reviewer #2: (No Response)

7. PLOS authors have the option to publish the peer review history of their article (what does this mean? ). If published, this will include your full peer review and any attached files.

**Do you want your identity to be public for this peer review?** For information about this choice, including consent withdrawal, please see our Privacy Policy .

Reviewer #2: No

---

## [Editor Report · Acceptance letter]

PONE-D-24-32665R2

PLOS ONE

Dear Dr. Harper, PE, PhD,

I'm pleased to inform you that your manuscript has been deemed suitable for publication in PLOS ONE. Congratulations! Your manuscript is now being handed over to our production team.

Kind regards,

on behalf of

Mr D. Daniel

Academic Editor

PLOS ONE